# Characterization of the Gut and Skin Microbiome over Time in Young Children with IgE-Mediated Food Allergy

**DOI:** 10.3390/nu16223942

**Published:** 2024-11-19

**Authors:** Michèle S. Roth, Muriel d’Aujourd’hui, Axel Künstner, Misa Hirose, Michael Olbrich, Saleh Ibrahim, Karin Hartmann, Caroline Roduit, Hauke Busch, Felicitas Bellutti Enders

**Affiliations:** 1Division of Pediatric Allergy, University Children’s Hospital Basel, 4031 Basel, Switzerlandfelicitas.belluttienders@ukbb.ch (F.B.E.); 2Institute for Experimental Dermatology, University of Lubeck, 23538 Lubeck, Germany; 3College of Medicine and Health Sciences, Khalifa University, Abu Dhabi 127788, United Arab Emirates; 4Division of Allergy, Department of Dermatology, University Hospital Basel and University of Basel, 4031 Basel, Switzerland; 5Department of Biomedicine, University Hospital Basel and University of Basel, 4031 Basel, Switzerland; 6Department of Clinical Research, University Hospital Basel and University of Basel, 4031 Basel, Switzerland; 7Children’s Hospital of Eastern Switzerland, 9000 St. Gallen, Switzerland; 8Division of Pediatric Respiratory Medicine and Allergology, Department of Paediatrics, Inselspital, University of Bern, 3010 Bern, Switzerland

**Keywords:** atopic dermatitis, fecal microbiome, immediate hypersensitivity, infant, longitudinal, microbiota, peanut allergy, pediatric, paediatric, tree nut allergy

## Abstract

Background/Objectives: The prevalence of food allergy (FA) in children is increasing. Dysbiosis of the microbiome has been linked to FA but needs to be better understood. We aimed to characterize the gut and skin microbiome of young food-allergic children over time and within different types of immunoglobulin E (IgE)-mediated FA. Methods: We studied 23 patients, as a pilot study of an ongoing prospective multicenter cohort study including children < 2y with newly diagnosed IgE-mediated FA. Samples (feces/skin swabs) were collected at enrollment and at 1-year follow-up and sequenced for the bacterial 16S rRNA gene (hypervariable v1–v2 region). Results: Gut and skin bacterial diversity was significantly higher in patients compared with controls and increased over time (beta test, Shannon diversity, *p* < 0.01). Within different types of IgE-mediated FA, bacterial diversity was similar. Community composition differed significantly over time and within IgE-mediated FA types (PERMANOVA: *p* < 0.01). Several significantly different genus abundances were revealed. We observed a positive correlation between high total IgE and a high abundance of the genus *Collinsella* in patients with a higher number of allergies/sensitizations (≥3), and patients with tree nut and/or peanut allergy. Conclusions: This study revealed an increased bacterial diversity in children with FA compared with non-atopic children. Importantly, the gut and skin microbiome differed in their composition over time and within different types of IgE-mediated FA. These findings contribute to the understanding of microbiome changes in children with FA and indicate the potential of the genus *Collinsella* as a biomarker for tree nut and/or peanut allergy and possibly for allergy persistence.

## 1. Introduction

The prevalence of food allergy (FA) is rising, affecting up to 8% of children worldwide [1,2,3]. Children with atopic dermatitis (AD) are at a higher risk of developing FA, especially those with onset in the first year of life [4,5]. Immunoglobulin-E (IgE)-mediated FA shows an individual natural course. Egg and milk allergies resolve in 70–80% of all cases, whereas tree nut and/or peanut allergies tend to persist, only resolving in about 20% [6,7,8,9,10]. Several risk factors are associated with FA persistence, mainly described in egg, milk, and peanut allergies, such as a larger wheal size in a skin prick test (SPT), higher specific IgE (sIgE), or a reaction to small amounts of allergen [6,9,11,12]. The early introduction of complementary feeding including allergenic foods in high-risk infants reduces the risk of developing FA [13,14]. Additionally, oral immunotherapies improve the course of FA by lowering the risk of severe allergic reactions, increasing the reaction threshold, and, in certain cases, achieving sustained unresponsiveness [15,16,17]. Nevertheless, biomarkers for FA persistence to improve prevention recommendations and to personalize treatments are not yet established.

The dual exposure hypothesis highlights the interplay between the gut, skin, and immune system to develop either allergy or tolerance to food allergens [18]. The microbiome plays an important role in educating and shaping the immune system in early life [19]. Early studies indicate that germ-free mice lack the ability to develop immunologically competent cells [20]. The microbial colonization stimulates the expansion of regulatory T-cells, favoring the development of tolerance to food allergens [21]. Dysbiosis is associated with atopic diseases, including FA [22,23,24]. Certain bacterial species, such as *Bifidobacterium adolescentis*, are considered protective against FA, whereas a lower abundance of short-chain fatty acid (SCFA)-producing bacteria, such as *Prevotella*, are associated with FA [25]. The transmission of the gut flora from non-allergic infants to allergy-prone mice resulted in reduced sensitization and allergy to food allergens [26]. Studies investigating the skin microbiome in children with AD observe an increased *Staphylococcus aureus* colonization in both lesional and non-lesional skin [27].

To date, no specific cluster of dysbiosis has been identified in individuals with FA. Few studies investigate the skin microbiome in children with IgE-mediated FA [28,29,30]. Several studies of the skin microbiome focus on children with AD [27,30,31,32]. The gut microbiome in children with FA is investigated in more detail, but mainly in older children and mostly with sampling at a single time point [25,33]. Furthermore, most studies include different types of IgE-mediated FA. Only a few studies focus on a specific type of IgE-mediated FA, e.g., egg, milk, or peanut allergy, and rarely include tree nut allergy [21,25,34,35,36,37].

As the gut, skin, and immune system are closely linked and the microbiome has a crucial impact on the immune system and thus on the development of allergy, it is important to characterize the gut and also the skin microbiome in young children with IgE-mediated FA. Different types of IgE-mediated FA show diverse clinical courses; thus, it is essential to describe their microbiome differences. Furthermore, as the microbiome is still developing in the first years of life, it is important to follow the microbiome changes over time. It is crucial to understand the microbiome’s impact on the course of FA to improve prevention and personalize treatment, possibly by identifying biomarkers.

Therefore, the objectives of our study are as follows: first, to investigate changes in gut and skin microbiome over time in children with IgE-mediated FA; second, to characterize the gut and skin microbiome of children with IgE-mediated tree nut and/or peanut allergy; and finally to identify biomarkers that can help predict the course of FA.

## 2. Materials and Methods

### 2.1. Study Design and Population

This study is a pilot study investigating 23 patients as a subset of a larger ongoing prospective multicenter cohort study designed to follow the clinical course of a high-risk group of children with IgE-mediated FA (Appendix A). Patients were recruited between July 2021 and 2022. At enrollment (denoted by t0) and 1-year follow-up (denoted by t1), medical history and diagnostic tests were performed. Recruitment took place at the Allergy Division of the University Children’s Hospital Basel (Switzerland) and at the Children’s Hospital of Eastern Switzerland, (Switzerland). Inclusion criteria were age < 2 years and diagnosis of IgE-mediated FA. Criteria for the diagnosis of an IgE-mediated FA were defined as a reliable history of an allergic reaction combined with the evidence of an IgE sensitization to the specific food allergen (positive SPT > 3 mm and/or sIgE ≥ 0.35 kU/l), and/or a positive result in an oral food challenge (OFC). These children are all at the stage of introducing new foods into their diet and families have followed the recommendations to introduce one new food at a time. This increases the accuracy of a reliable history. In the case of an unclear history or a positive IgE sensitization with unknown clinical relevance at baseline screening, further investigation was performed by OFC. A total of 44 OFCs were performed in 23 patients and FA was confirmed in 13 cases. Patients with IgE-mediated egg and/or milk allergy were started on an egg and/or milk ladder adapted from the Canadian protocol [38]. Briefly, a multistep protocol was used, starting with baked egg and/or milk and ending with raw food products. When egg and/or milk could be consumed without restriction, the allergy was considered resolved.

AD was diagnosed according to the Hanifin and Rajka criteria [39] and the severity was quantified using the SCORing Atopic Dermatitis (SCORAD) [40]. Patients were treated with a daily application of emollients. Topical corticosteroids were used intermittently as needed. None of the patients used probiotics or other oral supplements except the nationally recommended daily dose of vitamin D (400 IU/day).

The control group consisted of 8 children described by Reimer-Taschenbrecker et al. [41]. We selected controls that were best matched to the age of our patients. Controls were recruited by the pediatric dermatology clinic of the Department of Dermatology, University of Freiburg (Freiburg im Breisgau, Germany), between June 2018 and April 2020. Controls did not have FA or any atopic disease (AD, allergic rhinoconjunctivitis, or asthma), nor any other inflammatory skin diseases such as psoriasis. Skin swabs were taken from the forearm and feces samples were collected. The samples were processed the same way as in our cohort. All clinical and laboratory data were documented in SecuTrial^®^ (version 6.5.1.5, 2024).

### 2.2. Sample Collection

To investigate the gut microbiome, native feces were collected by parents at home using a provided feces sample collection kit (Stool Sample Collection and Stabilization Kit SC0011, Canvax Biotech S.L., Cordoba, Spain). Parents received verbal and written instructions. Samples were collected within 3–5 days after visits, kept at room temperature, mailed to the hospital, and then stored at −80 °C. This kit uses a DNA Stabilization Buffer to prevent DNA degradation and ensure nucleic acid stability. It allows storage at room temperature for up to 3 months, ensuring a stable microbiome. Feces samples were frozen within a median time of 6 days (t0 median 6 d (range 1–15), t1 median 7 d (range 1–15)).

For the skin microbiome, skin swabs were collected from two sites: the dorsal side of the forearm (referred to as “forearm”) and the antecubital fossa (“elbow”). These sites are considered to represent different predilection sites for AD. Inflammatory skin lesions are common in the antecubital fossa whereas the dorsal side of the forearm is often unaffected [42]. Skin sampling is described elsewhere [41]. Briefly, premoistened swabs were taken from the two sites and then stored at −80 °C on the same day.

### 2.3. Sequencing and Data Processing

For sequencing and analysis, samples were sent to the Lübeck Institute for Experimental Dermatology (LIED), Lübeck, Germany. Details about bacterial isolation, library preparation, sequencing, and sequence data processing are provided in the Appendix A.

### 2.4. Statistical Analysis

Alpha diversity was estimated on total abundances using the Shannon estimates from the DivNet package (v0.4.0) and heterogeneity of total diversity (observed plus unobserved) between conditions was assessed using the betta test (breakaway v4.8.4) with age as a fixed factor in addition to the factor investigated [43,44]. If necessary, patient ID was included as a random factor for repeated measurements. To estimate beta diversity, total abundance data were centered log-ratio (clr) transformed, and distances were calculated using Euclidean distance (Aitchison distance) [45]. Permutational multivariate ANOVA using distance matrices (permutational multivariate ANOVA) was used to analyze differences in beta diversity (adonis2 function, vegan package v2.6-4, with 9999 permutations) with patient ID as strata for constraint permutation. Pairwise adonis tests were performed using the package pairwise Adonis (v0.4). Differential taxa abundance analysis was performed on total abundance data using MaAsLin2 (v1.10.0) on TSS-normalized and arc-sine square root-transformed counts (as suggested in Nearing et al. [46,47]) for genera present in at least 20% of the samples with age as a fixed factor to adjust for age effects in the test. If necessary, patient ID was included as a random factor to adjust for repeated measurements. Correlations were calculated on relative abundances and Spearman’s ρ was used as correlation measurement.

Statistical analysis was performed using R (version 4.3.2). Visualizations of the microbiome data were performed using ggplot2 (version 3.4.4), ggpubr (version 0.6.0), and patchwork (version 1.2.0); the R packages phyloseq (version 1.44.0) and tidyverse (version 2.0.0) were used for data handling [48]; *p*-value and q-value were considered to be significant if <0.05, or otherwise stated.

## *3.* Results

### 3.1. Study Population Characteristics

The baseline characteristics are shown in Table 1. The median patient age at enrollment (t0) and 1-year follow-up (t1) was 12 and 24 months, respectively. The ages of patients at their 1-year follow-up and controls were similar (*p* = 0.8564). The details of atopic history, allergy characteristics, treatment, and clinical course are shown in Table 2. Tree nut and/or peanut allergy was present in fifteen patients (65%), of whom six were also diagnosed with egg and/or milk allergy. The remaining eight patients (35%) only had egg and/or milk allergy. Thirteen of the fourteen patients with egg and/or milk allergy had completed an egg and/or milk ladder, resulting in a tolerance of raw egg and/or milk. The patient who did not conduct an egg ladder remained allergic to egg. In the latest clinical follow-up in September 2024, 11 out of 15 patients (73.3%) with a tree nut and/or peanut allergy had persistent allergy. In four patients, the tree nut and/or peanut allergy resolved spontaneously with a tolerance of unrestricted allergen quantity.

### 3.2. Higher Diversity and Distinct Community Composition of Gut and Skin Microbiome in Patients with IgE-Mediated Food Allergy Compared with Controls

First, we compared the gut and skin microbiome of patients with controls. To evaluate differences in diversity we estimated sample-wise diversity (alpha diversity).

We observed a higher alpha diversity (Shannon) in patients’ feces samples at t1 and in patients’ forearm samples at t0 and t1 compared with controls (betta test; feces t1 *p* = 0.0048; forearm t0 and t1 *p* < 0.0001). The alpha diversity of feces samples at t0 was similar between patients and controls (betta test: t0 *p* = 0.4113) (Figure 1A, Appendix A). The Aitchinson distance was used to assess differences in community composition (beta diversity). In feces samples, PERMANOVA revealed a significant effect of group (control or patient) and sampling time point (t0 or t1) (*p* = 0.0001), whereas no significant effect of age on the community composition was detected (*p* = 0.1520). Community composition was significantly different when comparing feces samples (pairwise adonis test: *p* < 0.01, R2 = 0.06–0.08) and forearm samples (*p* = 0.01, R2 = 0.03–0.09) of patients at t0 and t1 with controls (Figure 1B, Appendix A).

The relative abundance analysis of the gut and the microbiome at phylum and genus level revealed a high heterogeneity between patients and controls, and between sampling time points and sampling sources (Figure 1C, Appendix A). Patients and controls showed similar dominant phyla. Firmicutes (also known as Bacillota) and Bacteroidota were the two dominant phyla in the gut microbiome. Actinobacteriota (also known as Actinomycetota) and Firmicutes were the dominant phyla in both patients, controls, and additionally *Proteobacteria* in the patients’ skin microbiome. At the genus level, *Faecalibacterium* and *Phocaeicola* were the two dominant genera in the gut microbiome. *Cutibacterium* and *Streptococcus* were the two dominant genera in patients’ forearm samples, and *Pseudoclavibacter* the was the dominant genus in patients’ elbow samples, whereas *Cutibacterium* and *Staphylococcus* were the two dominant genera in the skin microbiome of controls (Appendix A).

Comparing *clr*-transformed abundances at the genus level, we found a significantly increased abundance (q ≤ 0.05) of *Bacillus* (t1) in the gut microbiome and of *Alloprevotella* (t0, t1), *Arachnia* (t1), *Capnocytophaga* (t1), *F0422* (family *Veillonellceae*) (t0), *Fusobacterium* (t1), *Haemophilus A* (t1), *Lautropia* (t1), *Neisseria* (t0, t1), *Prophyromonas* (t0, t1), *Prevotella* (t0, t1), and *Pseudomonas E* (t0, t1) in the skin microbiome (forearm) in patients compared with controls. Further, we observed a decreased abundance (q ≤ 0.05) of *CAG 1427* (family *Eggerthellaceae*) (t0) in the gut microbiome and of *Gemella* (t0) and *Staphylococcus* (t0, t1) in the skin microbiome (forearm) of patients compared with controls (Figure 1D, Appendix A). Upon further investigation of the genus *Staphylococcus*, we found *Staphylococcus epidermidis* to be the dominant species in the skin microbiome (forearm) of patients and in controls. The identified *Staphylococcus* species all belong to the group of coagulase-negative *Staphylococci* (CoNS), and abundances were similar between patients at different time points and controls (Figure 1E and Appendix A). The species *Staphylococcus aureus* could not be detected due to its low abundance. However, it is debatable whether this species could not be detected due to its similarity to the other species.

### 3.3. Increasing Diversity and Changes In Gut and Skin Microbiome over Time Within Patients with IgE-Mediated Food Allergy

Next, we compared patients’ gut and skin microbiome at different sampling time points to assess changes over time.

The differences in alpha diversity and community composition are shown in Figure 2A,B. We found an increasing alpha diversity from t0 to t1 in both feces and the two skin sample sources (betta test: feces *p* < 0.0001, difference = 0.2509; forearm *p* < 0.0001, difference = 0.1274; elbow *p* < 0.0001, difference = 0.9421). The community composition was significantly different at the two sampling time points in feces (*p* = 0.0001) and forearm samples (*p* = 0.0028); no differences were detected for elbow samples (*p* = 0.0798). In elbow samples, we observed an age effect (*p* = 0.0102), whereas feces and forearm samples showed no effect of age on community composition (*p* > 0.1).

Comparing the abundances at the genus level, we found, from sampling time point t0 to t1, an increase in abundances (q ≤ 0.05) of *Bacillus*, *Faecalibacterium*, *Pseudomonas*, and *Sellimonas* in the gut microbiome and of *Abiotrophia* (elbow and forearm), *Actinomyces* (forearm), *Arachnia* (forearm), *Kingella A* (forearm), Knoellia (forearm), *Lautropia* (forearm), *Neisseria* (forearm), and *Lactococcus A* (elbow) in the skin microbiome. We observed a significant decrease in abundances (q ≤ 0.05) of *Pseudoclavibacte*, and *Veillonella* in the gut microbiome, and *Pseudoxanthomonas* (forearm) in the skin microbiome over time from t0 to t1 (Figure 2C–E).

### 3.4. Similar Diversity in Gut and Skin Microbiome and Distinct Community Composition in Gut Microbiome of Patients with Different Types of IgE-Mediated Food Allergies

Third, we compared the gut and skin microbiome at enrollment of patients with IgE-mediated tree nut and/or peanut allergy (n = 15) with patients with an IgE-mediated FA other than tree nut and/or peanut (n = 8) to identify differences within different types of IgE-mediated FA.

The alpha diversity of the gut and skin microbiome (forearm and elbow) was similar in patients with different types of IgE-medaited FA (betta test; feces *p* = 0.970; forearm *p* = 0.0538; elbow *p* = 0.7002) (Figure 3A). Community composition significantly differed in the gut microbiome, but not in the skin microbiome of patients with tree nut and/or peanut allergy compared with patients with other types of FA, with an age effect observed in feces and elbow samples (nut vs. no nut: feces *p* = 0.0114; forearm *p* = 0.2365; elbow *p* = 0.9783) (age: feces *p* = 0.0070; forearm *p* = 0.1231; elbow *p* = 0.0039) (Figure 3B).

Comparing the abundances at genus, we found significantly increased abundances (*p* < 0.05) of *Blautia A*, *Enterocloster*, *Lawsonella*, and *Phocaeicola* in the gut microbiome and of *Cutibacterium* (forearm) and *Sphinogomonas* (elbow) in the skin microbiome of patients with tree nut and/or peanut allergy. We observed significantly decreased abundances (*p* < 0.05) of *Enterococcus D*, *Neisseria*, *Pseudomonas E,* and *Veillonella* in the gut microbiome and of *Gemella*, *Granulicatella*, *Haemophilus D*, *Lancefieldella, Nanogingivalis*, *Nanosynbacter*, *Pauljensenia*, and *Veillonella* in the skin microbiome (forearm) of patients with tree nut and/or peanut allergy (nut) compared with patients with FA other than tree and/or peanut (no nut) (Figure 3C–E).

### 3.5. Correlation of the Genus Collinsella with High Total IgE in Patients with Increased Numbers of Allergies/Sensitizations

Further, we investigated whether total IgE levels and the number of allergies/sensitizations to food allergens correlated with genera abundances at enrollment using Spearman’s rank correlation.

We observed that patients with high total IgE (>100 kIU/mL) show more allergies/sensitizations (Fisher’s exact test: *p* = 0.0499). High total IgE positively correlated with a high abundance of the genus *Collinsella* in the gut and skin microbiome when stratifying according to number of allergies/sensitization (≥3) (feces *p* = 0.0052; elbow *p* = 0.0049, forearm *p* = 0.0432). Interestingly, this was the only genus with a significant correlation in all three sample sources (Figure 4). Stratifying only according to the number of allergies (without the number of sensitization), we found an increased abundance of the genus *Collinsella* in the skin microbiome, but not in the gut microbiome (feces *p* > 0.05, forearm *p* = 0.041, elbow = 0.022). However, stratifying according to the type of FA, we found a significant correlation between high total IgE and the abundance of the genus *Collinsella* in the feces and elbow samples of patients with tree nut and/or peanut allergy, but not in forearm samples (feces *p* = 0.001, elbow = 0.0418, forearm = 0.1296) (Appendix A).

## 4. Discussion

As the prevalence of FA increases, the ability to personalize treatment and tailor preventions becomes more important. The microbiome is an important player in the development of FA. To better understand this interaction, we investigated the gut and skin microbiome in a high-risk population of young children with newly diagnosed IgE-mediated FA. We would like to emphasize that we did not only investigate the gut, but also the skin microbiome, and we followed the microbiome’s evolvement over time and differentiated between different types of IgE-mediated FA. We identified higher bacterial diversity in patients with FA and an increase in diversity over time. Furthermore, we determined a distinct community composition in our patient, with shifts in the gut and skin microbiome over time and within different types of IgE-mediated FA. *Collinsella* emerged as a noteworthy genus, as its abundance in the gut and skin microbiome is correlated with high total IgE in patients with increased numbers of allergies/sensitizations to food allergens and, particularly, is elevated in patients with tree nut and/or peanut allergy.

The alpha diversity of our patients’ gut and skin microbiomes increased over time. Few longitudinal studies have documented these changes in children with FA and AD. Kennedy et al. reported an increase in the skin microbiome diversity (Shannon index) from 2 to 6 months in children with AD [49]. Another study observed, after an initial decrease from birth to 6 months, a continuous increase in the alpha diversity (Sobs, Shannon, and Simpson indices) of the gut microbiome of both children who developed AD at two years and controls [50]. Similarly, Galazzo et al. found an increase in the gut microbiome diversity (Shannon index) after 31 weeks, linking it to atopic diseases [51]. These findings indicate that the microbiome development in children with FA and AD is similar to those without these conditions [52,53,54]. Furthermore, the bacterial diversity of both microbiome sources was similar in patients with different types of IgE-mediated FA. Chun et al. noted shifts in the gut microbiome diversity (Simpson index) in food-allergic children who later developed peanut allergy, with lower diversity in infancy but similar diversity by mid-childhood [55]. Our observations suggest shared underlying pathophysiological mechanisms within FA, despite the individual clinical courses, highlighting the importance of conducting longitudinal studies to reveal similarities and differences. Interestingly, our patients’ gut and skin microbiome showed higher bacterial diversity than the controls. It is well known that the microbiome is influenced by multiple factors. Regarding the gut microbiome, several studies have reported lower diversity in children with FA [25,33,35,56], but other studies have found no effect [29,57,58,59]. Berni Canani et al. and Fazlollahi et al. are in line with our findings, showing higher gut microbiome diversity in children with egg and milk allergies [34,60]. One explanation for the high diversity of our patients’ gut microbiome might be the early introduction of a diverse diet, which was encouraged by dietary counseling. A broad diet is associated with a favorable effect on the microbiome [61,62]. Regarding the skin microbiome, our patients with mild AD regularly applied emollient, which is suggested to improve the skin condition. Previous studies comparing the effects of an emollient on the skin microbiome in children with the risk of developing AD or having a diagnosis of AD showed an increased diversity of the skin microbiome when treated with emollients [63,64,65].

Like previous studies, we observed high heterogeneity between the samples, reflecting the evolvement of the microbiome in infants [49,51,54]. This age period has been hypothesized to be the window for shaping the immune system and allowing the development of tolerance to food allergens, thus making our analysis at this age significant [19]. Despite the heterogeneity, taxonomic abundance analysis revealed the most dominant phyla and a distinct community composition with significantly different genus abundances in our patients. Firmicutes and Bacteroidota were the two dominant phyla in the gut microbiome of both patients and controls, which is related to the introduction of solid food [66].

At the genus level, the shifts in the gut microbiome over time of several genera of the phylum *Firmicutes* are noteworthy. While *Faecalibacterium*, *Bacillus*, and *Sellimonas* increased, *Veillonella* decreased over time. This is in line with a study evaluating the gut microbiome of children with food sensitization at 13 months of age [33]. Similarly, Filippis et al., studying children (5 years) with respiratory allergy and FA, and Reddel et al., investigating toddlers (2 years) with AD, observed an increased abundance of *Faecalibacterium* in the gut microbiome [36,67]. A decreased abundance of *Veillonella* is observed already at the age of 2 months in the gut microbiome of Japanese children who developed FA within the first two years of life [68]. Fan et al. described a lower abundance of *Veillonella* in the gut microbiome of children with AD at 2 years of age [50]. Our observations, together with these studies, suggest that we have revealed important changes in the microbiome of young children with FA.

In our study, we correlated the abundance of the genus *Collinsella* in the gut and skin microbiome with high total IgE in patients with three or more allergies/sensitizations to food allergens. Further, we observed an increased abundance of the genus *Collinsella* in the gut and skin microbiome in patients with tree nut and/or peanut allergy, which are allergies that tend to persist. Consistent with our findings, the genus *Collinsella* is described as one of thirteen dominant genera in children with food sensitization [33]. Moreover, Goldberg et al. observe an increased abundance of *Collinsella aerofaciens* in children with FA [25]. *Collinsella* (phylum Actinobacteriota) is described to produce short-chain-fatty acids (SCFA), specifically butyric acid, which was shown to have a possible protective effect against allergic diseases [69,70]. Hence, we would expect a low abundance of *Collinsella* in patients with tree nut and/or peanut allergy, which contradicts our results. However, studies with a bigger sample size of tree nut and peanut allergic patients and investigations of the metabolic function of this genus would be required to evaluate whether *Collinsella* may serve as a biomarker for tree nut and/or peanut allergy and thus allergy persistence.

Previous studies have focused on *Staphylococcus aureus* and AD, limiting the interpretation of the skin microbiome results [71]. More recently, the diversity of the skin microbiome has been better recognized, but few studies examine the skin microbiome in children with FA [28,29]. The dominant phyla in the skin microbiome of our patients and controls were Actinobacteriota and Firmicutes, which is in line with previous studies [54,72]. Our findings showed a lower abundance of the genus *Staphylococcus* compared with controls. These were mainly *Staphylococcus* species belonging to the coagulase-negative *Staphylococci* (CoNS) with *S. epidermidis* as the most dominant one. Our patients had mild AD, which was represented by the low SCORAD, as a basic treatment with emollients was applied regularly. Byrd et al., comparing the skin microbiome composition at different states of pediatric AD, showed, similarly to our results, a predominance of *S. epidermidis* in less severe disease and at post-flare state, but is not significant in the latter [27]. Furthermore, the genus *Staphylococcus* is lower in the skin microbiome of infants who were later affected by AD, emphasizing a protective role of *Staphylococcus*, mainly CoNS, in the development of skin diseases [49]. Following the microbiome over time and investigating different types of IgE-mediated FA, we found no changes in the genus *Staphylococcus*.

## 5. Limitations

We recognize that there are limitations to this study. The small sample size limits the generalization of our results. This is a study with young children, and the availability of controls was challenging; therefore, we included only eight controls without any allergies. As this is an ongoing cohort study, we could provide, to date, only data from two sampling time points, and the evaluation of allergy persistence is pending. Furthermore, the analysis of the metabolic function of a specific bacterium, e.g., *Collinsella* in a murine or an organoid model, would provide further insights into the host–microbiome interaction.

## 6. Conclusions

This cohort study provides an extension of the few studies investigating the gut and especially the skin microbiome in young children with FA. Additionally, the study analyzes the microbiome longitudinally and includes the description of the microbiome of different types of IgE-mediated FA. By characterizing the gut and skin microbiome, we found an increase in bacterial diversity in young children with FA and over time. Importantly, the gut and skin microbiome differed in community composition over time and within different IgE-mediated FA. This indicates that the microbiome evolves with the host over time and supports the hypothesis that underlying pathomechanisms within different IgE-mediated FA may vary according to their different clinical courses. Further studies with longer observation periods are needed to better understand the dynamics of the gut and skin microbiome over time. In addition, to gain insight into the interaction between the host and the microbiome, the integration of the metabolome is desirable.

## Figures and Tables

**Figure 1 nutrients-16-03942-f001:**
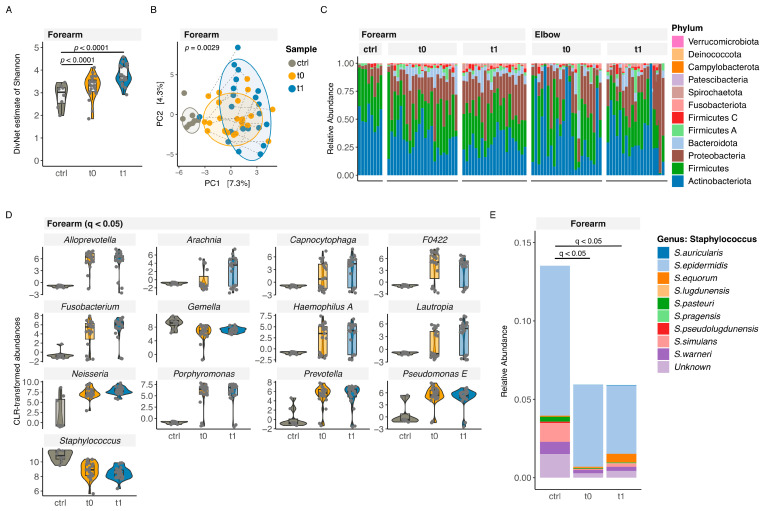
Diversity and community composition of the skin microbiome of patients with IgE-mediated food allergy (FA) compared with controls. (**A**) Violin plot of alpha diversity (Shannon estimate; betta test) and (**B**) RDA (redundancy analysis) plot of community composition (beta diversity) of forearm samples of controls and of patients at two different sampling time points (ctrl, sample of control; t0, sample of patients at enrollment; t1, sample of patient at 1-year follow-up; PC, principal coordinate; each dot represents one sample; gray dot, sample of control; yellow dot, sample of patient at enrollment (t0), blue dot, sample of patient at 1-year follow-up (t1); colored ellipses comprise 80% of the data). (**C**) Panel of relative abundances at phylum level of individual samples of skin of patients and controls. Violin plot of differences in clr-transformed genus abundances of forearm (**D**). (**E**) Panel of the relative abundance of the genus *Staphylococcus* at species level of forearm samples. (forearm/elbow t0 n = 23, t1 n = 21; significance level as indicated in figures).

**Figure 2 nutrients-16-03942-f002:**
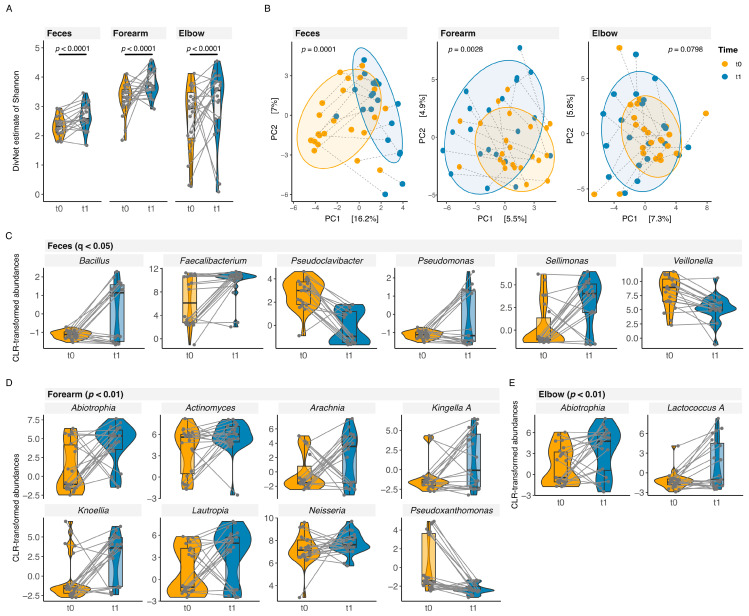
Diversity and community composition over time of the gut and skin microbiome of patients with IgE-mediated food allergy (FA). (**A**) Violin plot of alpha diversity (Shannon estimate; betta test) and (**B**) RDA (Redundancy Analysis) plot of community composition (beta diversity) of feces, forearm, and elbow samples at different sampling time points (PC, principal coordinate; each dot represents one sample of a different sampling time point: yellow dot, sample at enrollment (t0); blue dot, sample at 1-year follow-up (t1); colored ellipses comprise 80% of the data). Violin plot of differences in clr-transformed genus abundances at two different sampling time points of feces (**C**), forearm (**D**), and elbow samples (**E**). (feces t0 n = 22, t1 n = 20; forearm/elbow t0 n = 23, t1 n = 21; significance level as indicated in figures).

**Figure 3 nutrients-16-03942-f003:**
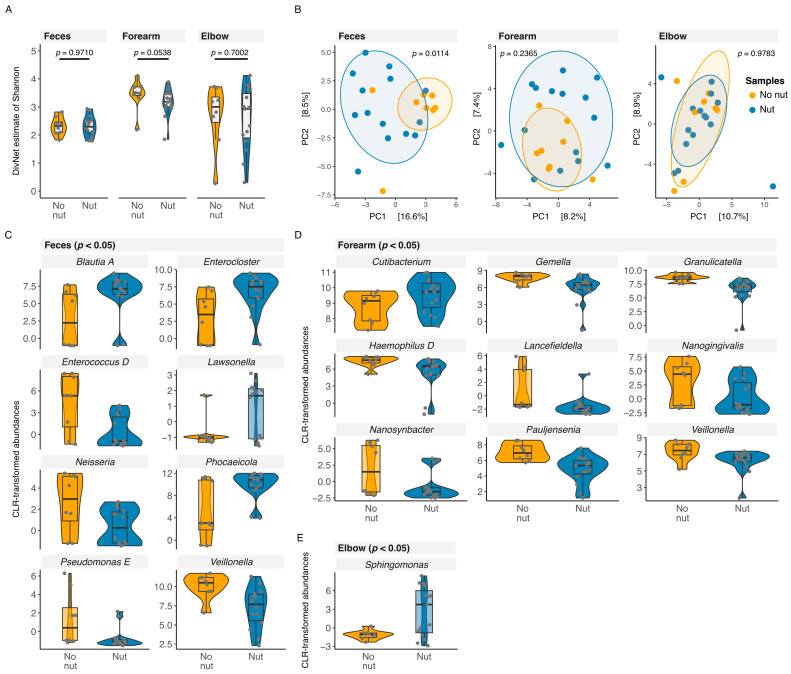
Diversity and community composition of the gut and skin microbiome of patients with different types of IgE-mediated food allergies (FA). (**A**) Violin plot of alpha diversity (Shannon estimate, betta test) and (**B**) RDA (redundancy analysis) plot of community composition (beta diversity) of feces, forearm, and elbow samples. (PC, principal coordinate; colored ellipses comprise 80% of data; each dot represents one sample: yellow dot, sample of patient with an IgE-mediated FA other than tree nut/peanut allergy; blue dot, sample of patient with a tree nut and/or peanut allergy). Violin plot of differences in clr-transformed genus abundances of feces (**C**), forearm (**D**), and elbow (**E**) samples. (Significance level as indicated in figures; investigated time point t0; no nut, sample of patient with an IgE-mediated FA other than tree nut and/or peanut, n = 8; nut, sample of patient with a tree nut and/or peanut allergy, n = 15).

**Figure 4 nutrients-16-03942-f004:**
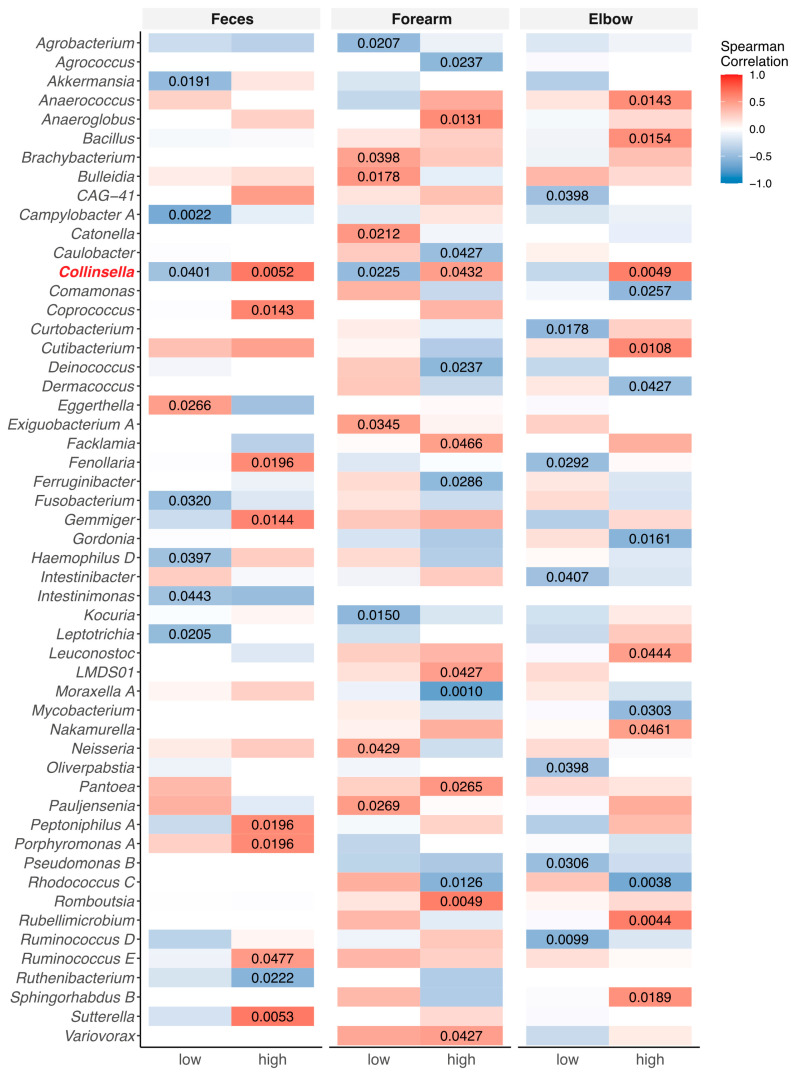
Correlation of gut and skin microbiome of patients with IgE-mediated food allergy (FA) with allergy parameters. Spearman’s correlation of genera abundances of feces, forearm, and elbow samples with total IgE, stratified according to number of allergies/sensitizations (*x*-axis: number of allergies/sensitization; low ≤ 2; high ≥ 3; only significant *p*-values are shown (*p* < 0.05); red, positive correlation; blue, negative correlation; investigated time point t0).

**Table 1 nutrients-16-03942-t001:** Demographic characteristics of patients grouped according to type of IgE-mediated food allergy (FA) at enrollment and of controls.

	Patients			Controls n = 8	*p*-Value (All vs. Controls)
	All n = 23	Nut n = 15	No Nut n = 8		
sex (female)	7 (30%)	4	3	4 (50%)	0.4055
age (month)	12 (3–21)	13.7 (3–14)	10.5 (7–18)	26 (3–46)	0.0237
delivery mode					0.3931
vaginal	18 (78%)	14	5	5 (63%)	
Cesarian section	5 (22%)	0	3	3 (38%)	
birth weight (g)	3325 (1845–4850)	3399 (2900–3940)	3437 (1845–4850)	na	
gestational age (weeks)	40 (34–42)	40 (37–42)	40 (34–41)	na	
breast feeding	21 (95%) ^§^	14	7	na	
age of food introduction (month)	5 (4–6) ^ç^	5(4–6)	4.7 (4–5)	na	
sibling(s)	12 (71%) °	12	3	na	
living condition					0.6417
urban	16 (70%)	9	7	7 (88%)	
rural	7 (30%)	6	4	1 (13%)	
day care	3 (23%) ^$^	2	1	na	
pets	3 (13%)	3	0	1 (13%)	1.0000
sampling of microbiome					0.0736
season (winter/spring/summer/autumn)	4/9/5/5	3/5/4/3	1/4/1/2	1/1/0/6	
skin					
forearm	23	15	8	8	
elbow	23	15	8	na	
feces	22	14	8	8	

Fisher’s exact test was used for categorial variables and the Wilcoxon test for continuous variables. Data are shown as median (range) or numbers (%). *p*-value comparing “all patients” with “controls”. “nut”-group includes patients with tree nut and/or peanut allergy; “no nut”-group includes patients with IgE-mediated FA other than tree nut and/or peanut. Deviation of sample size: § n = 22; ° n = 17; ç n = 14; $ n = 13. Abbreviations: IgE, immunoglobulin E; na, not available.

**Table 2 nutrients-16-03942-t002:** Patients’ atopic history, allergy characteristics at enrollment (t0), treatment, and clinical course.

	Patients			*p*-Value (Nut vs. No Nut)
	All n = 23	Nut n = 15	No Nut n = 8	
**Atopic history**				
atopic dermatitis	21 (91%)	14	7	
SCORAD	7.04 (0–24)	8 (0–24)	5 (0–8.8)	0.4711
allergic rhinoconjunctivitis	1 (4%)	1	0	
multiple trigger wheezing	1 (4%)	0	1	
atopic family history	19 (83%)	12	7	
**Allergy characteristics**				
total IgE (kIU/mL)	36.5 (1.48–1230) *	211.96 (9–1230)	69.35 (1.48–356)	0.0569
number of allergies/sensitizations (sensitizations with and without clinical relevance)	2 (1–8)	4 (1–8)	1 (1–2)	0.0457
low: ≤2 total sensitizations	12	4	8	
high: ≥3 total sensitizations	11	11	0	
number of allergies (range)	2 (1–4)	2 (1–4)	1 (1–2)	
allergy to …				
egg	11 (48%)			
milk	6 (26%)			
tree nut	12 (52%)			
cashew nut	7			
hazelnut	5			
walnut	3			
peanut	7 (30%)			
allergy distribution				
only egg and/or milk allergy	8 (35%)			
only tree nut and/or peanut allergy	9 (39%)			
mixed allergy to egg, milk, tree nut and/or peanut	6 (26%)			

Fisher’s exact test was used for categorical variables and the Wilcoxon test for continuous variables. Data are shown as median (range) or numbers (%). *p*-value for comparison of “nut” and “no nut”-groups; “nut”-group includes patients with tree nut and/or peanut allergy; “no nut”-group includes patients with IgE-mediated FA other than tree nut and/or peanut. Deviation of sample size: * n = 20. Abbreviations: IgE, immunoglobulin E; SCORAD, SCORing Atopic Dermatitis.

## Data Availability

All related data and materials are published in this manuscript. Patient sequencing data (fastq files) have been deposited in the European Nucleotide Archive (ENA) under accession number PRJEB61516. For the controls, data (fastq files) were retrieved from the European Nucleotide Archive accession number PRJEB47047.

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
