# Peer review of "Characterization of the Gut and Skin Microbiome over Time in Young Children with IgE-Mediated Food Allergy"

_nutrients, 2024, doi:10.3390/nu16223942_

Round 1
Reviewer 1 Report
Comments and Suggestions for Authors
Roth et al. presented the results of gut and skin microbiome analyses among 23 children who suffered from food allergy (FA) identified from an ongoing multicenter cohort study in Europe. Stool and skin swab samples were subjected to 16S rRNA sequencing. They found some changes in microbial diversity and compositions at both gut and skin in young children who were sampled at baseline (t0) and 12 months later (t1). While the results are clinically relevant and potentially impactful, the authors need to address a number of issues to improve this manuscript.
First, I am concerned how the 23 studied subjects were identified from the entire cohort. Were all subjects from the entire cohort recruited into the present microbiome study? If not, what were the recruitment strategy and criteria in this study? Were the characteristics of these 23 subjects representative of all FA cases in the cohort as well as the entire cohort population? Can there be significant biases in the present microbiome analyses? Many large-scale studies on the epidemiology of FA suggested a figure of 3-5% FA in early childhood, so I would expect more FA cases than 23 subjects to be found in this multicenter cohort.
How were the eight controls identified? Why did they recruit only 8 controls which was far fewer than the number of FA cases? Did they perform any sample size calculation? I would expect similar numbers for FA cases and controls in this study, which would offer higher study power.
My second question relates to the diagnosis of FA. As described on lines 95-98, FA was diagnosed either by “reliable history” plus evidence of IgE sensitization by SPT/sIgE or by an oral food challenge (OFC). There is some degree of heterogeneity in the FA definition. What were the “reliable” history? Even at the best reporting, the former approach of FA diagnosis would overestimate the incidence with some false-positive cases. How many of the 23 studied subjects were diagnosed by history plus IgE sensitization and by OFC (Table 2)? Were the patient characteristics and microbiome-related data different in FA patients diagnosed by these two approaches?
My another concern relates to lack of data on subjects’ dietary intake. Food restriction was expected among young children with FA, especially to milk and egg. What were the possible effects of this altered dietary practice on gut (and ? skin) microbiome? Some patients with milk and egg allergies were advised to follow the respective ladders so the their diet also changed during the 12-month follow-up. How would this influence gut microbiome? Could the increase in diversity and changes in microbial compositions at t1 reflect these dietary changes? It seemed from the data at t0 that gut microbiome was similar between FA case and controls.
Regarding the skin sampling for microbiome analyses, the authors collected skin swab samples from dorsal side of forearm and antecubital fossa. Clinical experience suggests that eczema commonly affects the latter body site. How would the authors proceed with skin sampling in such situation? Did they give up to collect skin swabs from antecubital fossa? If they proceeded to collect skin swabs from antecubital fossa affected by eczema, we would expect altered skin microbiome in these affected sites when compared even with unaffected body sites of the same children.
This study did not report the use of additional therapies in relation to eczema or FA. For example, were subjects standardized regarding the use of emollient and topical corticosteroids? If not, how would such confounders influence skin microbiome? Similar for FA, did children take oral supplements or probiotics/prebiotics? How would such dietary intakes affect gut microbiome? It is important for the authors to present more detailed data on the dietary intakes and supplements as well as eczema-related treatment in this study.
Comments on the Quality of English LanguageReasonable and no major concern
Author Response
We thank you for your comments. Please see the attachment.

Reviewer 2 Report
Comments and Suggestions for Authors
This is well-written, prepared and designed research. I have some concerns which could help improve the quality of the paper, I hope:
- more information about the control group can be added - criteria of in/exclusion and the place of recruitment/type of recruitment
- did the required sample size be calculated? If not, Authors could assess the power of the test
If the number of individuals was not estimated, is this the pilot study or preliminary results, i supposed. Do the authors agree? If yes, this information should be added to the title/method/abstract, etc.
- How long were the stool samples kept at room temperature after collection? Please add information about maximum time - this time, if any stabilization was used, could highly affect the results of the microbiome analysis
- in Table 1, the information on which groups are compared where the p is estimated in a more visible way, and also the multiple comparisons between nut/no nut/control should be determined
- It's important to add a separate section on the strength and limitations of the study. This will provide a clear understanding of what the study has achieved and where it could be improved, enhancing the overall quality of the manuscript.
Some less important issues:
- Authors should carefully revise the manuscript and adjust them according to the Nutrients guidelines, f.e. structure of the Abstract (headings)
- I highly recommend adding the graphical version of the Abstract with the most critical findings and information about the study group. The study is well written, has the potential impact and soundness and may arouse great interest
Author Response

(The authors gave the same response as above.)
